# Changes in risk habits and influencing factors in the Taiwan oral cancer screening program

Pattaranan Munpolsri[1], Chiu-Wen Su[2], Hsu-Fei Yang[3�], Tsui-Hsia Hsu[3ͦ], Yen-Yu Chou[3ͦ], Li-Ju Lin[3ͦ], Chao-Chun Wu[3ͦ], Sam Li-Sheng Chen[ID][4], Amy Ming-Fang Yen[4]*

1 School of Dentistry, College of Oral Medicine, Taipei Medical University, Taipei, Taiwan, 2 Department of Internal Medicine, National Taiwan University Hospital, Taipei, Taiwan, 3 Health Promotion Administration, Ministry of Health and Welfare, Taipei, Taiwan, 4 School of Oral Hygiene, College of Oral Medicine, Taipei Medical University, Taipei, Taiwan

☉ These authors contributed equally to this work.
* amyyen@tmu.edu.tw

## Abstract

This study examines changes in oral risk habits and identifies factors influencing these changes among participants in a population-based oral cancer screening program to support effective public health interventions. The study included 2,569,920 individuals aged 30 and older who participated in Taiwan's Oral Cancer Screening Program at least twice between 2010 and 2021. Changes in cigarette smoking and betel quid chewing were assessed between the first and last screenings and categorized as improved, unchanged, or worsened. A logistic regression model evaluated factors associated with habit improvement, including baseline oral habits, sex, age, education, screening adherence, and oral potentially malignant disorder (OPMD) findings. Among participants, 25.3% improved their oral habits. Baseline habits influenced how OPMD screening results affected behavior change. Among smokers, a positive screening result increased the likelihood of quitting or reducing smoking (adjusted odds ratio [aOR] = 1.18, 95% CI 1.16–1.20). However, among betel quid chewers, whether or not they smoked, a positive screening result was negatively associated with improved habits (aOR 0.79–0.88). Being female, older, college-educated, and regularly attending screenings were positively linked to behavior improvement. The program led to habit improvements in about one-quarter of participants, particularly older individuals, those with higher education, and frequent attendees. However, a diagnosis of OPMD motivated change only among smokers, not those engaging in both smoking and betel quid chewing, highlighting a lack of awareness in high-risk groups. Strengthening collaboration between health organizations and the screening program could enhance public awareness, improve program effectiveness, reduce oral cancer incidence, and lower long-term healthcare costs.

**Data availability statement:** The aggregated dataset used in this study has been publicly uploaded to the Dryad repository and is available at: https://doi.org/10.5061/dryad.612jm64h3. or http://datadryad.org/share/_Y6sTNL-JPFygiwCL95Q9EZaCD_wVZ-BrRlSoGUC64_A

**Funding:** This study was supported by the Health Promotion Administration, Ministry of Health and Welfare (A1111113), and the National Science and Technology Council, Taiwan (113-2314-B-038-085-MY3). The funds will be received and managed by AMFY. The funding sources had no role in the study design, data collection, analysis, interpretation, manuscript writing, or the decision to submit the paper for publication.

**Competing interests:** The authors have declared that no competing interests exist.

## Introduction

Oral well-being is an essential part of overall health, and preventive measures are crucial in reducing the risks associated with harmful oral habits. Risky oral habits such as smoking and betel quid chewing are major factors contributing to oral health problems, including an increased risk of oral cancer and related diseases.

Oral cancer is one of the most dangerous diseases among all oral conditions, caused by the abnormal growth of cells in the oral cavity. Several factors contribute to the development of oral cancer, including smoking, betel quid chewing, alcohol consumption, chronic inflammation, genetics, and immunodeficiency. Treatment options for oral cancer include surgery, radiation therapy, chemotherapy, or a combination of these methods. The success rate of treatment decreases with the progression of the disease, so early detection of oral cancer and the avoidance of risk habits are essential [1–16].

Oral cancer screening programs can significantly reduce mortality rates by enabling early detection of oral cancer and oral potentially malignant disorders (OPMDs), allowing timely referral for appropriate treatment [17–19]. These screening programs recognize the importance of early detection and intervention, making them a vital tool for promoting oral health awareness and encouraging habit change [20]. Participants in such screening programs are expected to modify their habits appropriately to reduce their risk of developing oral diseases in the future.

Previous research, whether based on surveys or cohort databases, suggests that factors such as dosage of consumption, educational level, illness, and intention to adjust habits have an impact on the likelihood of success in promoting positive changes in tobacco and betel quid chewing habits. Nevertheless, these investigations are limited to enhancing a single harmful habit, such as smoking or betel quid chewing [21–27]. However, in reality, most individuals with oral risk habits do not engage in just one single risky habit, as these habits can often trigger or reinforce the development of other associated risk habits [28].

This study aims to report the changes in oral risk habits among individuals participating in an oral screening program that targeted those engaged in high-risk oral habits, including cigarette smoking and betel quid chewing. We also investigated factors associated with changes in oral habits, yielding more precise insights into the consequences of harmful modifications. The primary objective of this study is to investigate the changes in oral risk habits and assess the factors linked to the adoption of advantageous oral habits among individuals participating in a population-based oral cancer screening program in Taiwan. This analysis will consider combining two specific oral risk habits, cigarette smoking and betel quid chewing.

## Materials and methods

### Study design and setting

This is a retrospective cohort study that uses data from the oral cancer screening program in Taiwan.

## Data source and collection

This study utilized data collected between 2010 and 2021 from a population-based oral cancer screening program. Individuals aged 18 and above who smoke or chew betel quid are voluntarily invited to attend the biennial screening program, which is reimbursed by the Health Promotion Administration (HPA). All screening procedures were conducted at local clinics and hospitals. Oral examinations were performed by dentists, otolaryngologists, and other specialists who had completed a training program provided by HPA. This training, conducted by certified oral surgeons and senior dentists, lasted approximately three days and included both theoretical instruction and practical assessments. The program was specifically designed to enhance the detection of suspicious oral mucosal lesions. Suspected cases identified during the screenings were referred to hospital specialists for confirmatory pathological diagnosis. A screening data monitoring and surveillance center recorded screening, referral, and diagnosis data. Demographic information, oral habits, and screening results were collected at each visit. A structured questionnaire was used to gather demographic data and information on cigarette smoking and betel quid chewing through face-to-face interviews conducted in community and hospital settings [19]. Trained dentists or physicians conducted visual examinations of participants' oral cavities to identify Oral Potentially Malignant Disorders (OPMDs), including oral leukoplakia, erythroleukoplakia, erythroplakia, oral submucous fibrosis, and verrucous hyperplasia. Chuang (2017) outlined the coding issues and specifics of this screening program [19]. Health education, including information on risk factors, risky behaviors, and proper oral hygiene practices, was provided immediately after the screening. Each session typically lasted 10–15 minutes per participant and included verbal counseling along with visual aids such as pamphlets and posters. Public health nurses and trained educators delivered the education under the supervision of dental professionals.

It is recognized that interview-based data collection is prone to biases, including interviewer bias, social desirability bias, and recall bias. To minimize these biases, all interviews were conducted by trained professionals as mentioned above. For study purposes, data from January 2010 to December 2021 were accessed on November 22, 2022. Information that could identify individual participants during or after data collection was accessible only to Dr. Amy Ming-Fang Yen, the co-principal investigator of the National Oral Cancer Screening Surveillance Program. The aggregated dataset used in this study has been publicly uploaded to the Dryad repository and is available at: https://doi.org/10.5061/dryad.612jm64h3.

## Participants and variables

This study includes all screening records that met specific criteria: participants aged over 30 who attended the oral cancer screening program in Taiwan at least twice between 2010 and 2021. Based on findings from a previous study indicating that individuals under the age of 30 have a very low risk of malignancy, this study set the inclusion criterion at age over 30 [20]. After excluding 10,853 individuals under the age of 30, data from approximately 2.6 million participants were included in the analysis. The dataset contains information on oral habit changes, age groups (30–45, 46–60, 60+), sex, education level (elementary school, middle and high school, college or higher), screening frequency (only twice, more than twice), and oral potentially malignant disorders (OPMDs) status (positive or negative findings). As part of the screening program, oral cancer risk and prevention education were provided, and changes in oral habits were recorded at each screening to evaluate the effectiveness of the educational component.

## Measurement of cigarette smoking and betel quid chewing

This study intends to investigate two habits, cigarette smoking and betel quid chewing. This study used 10 years of use and 20 pieces per day as the criteria for classifying levels of oral habits, based on previous studies [29,30]. The use of cigarette smoking is categorized into 3 levels: LS: never smoke, cessation, and low degree (<10 years and <20 pieces per day); MS: medium degree (<10 years and ≥20 pieces per day, ≥ 10 years and < 20 pieces per day); and HS: high degree (≥10 years and ≥20 pieces per day). The betel quid chewing is categorized into 5 levels NB: never chewing, QB: cessation, LB: low degree (<10 years and <20 pieces per day), MB: medium degree (<10 years and ≥20 pieces per

day, ≥ 10 years and < 20 pieces per day), and HB: high degree (≥10 years and ≥20 pieces per day). The combined oral habit changes between the first and the last screening for cigarette smoking and betel quid chewing were rated to improve and not improve (Fig 1a). The improvement of the combination of smoking and betel quid chewing indicates that at least 1 habit is being improved with no other habit worsening (Fig 1d). The improvement of smoking and betel quid chewing was defined as shown in Fig 1b, c.

## Statistical analysis

The oral habit changes (improved, no change, or deterioration) were summarized as numbers and percentages by the following characteristics: oral habits of the first screening, sex, age, education, screening times, and OPMD screening results. The investigation of the associated factors for improving habit changes was established with the logistic regression model and 95% confidence intervals (Wald method). The key assumptions for logistic regression were met, including independence of observations, a binary outcome, and absence of multicollinearity. The data were collected from independent observation, and all variables showed acceptable variance inflation factor (VIF) values (<2), indicating no serious multicollinearity. The outcome of the model was improvement or non-improvement. For missing data, this study categorized it as one level of the data, allowing for adjustments during the analysis. Note that individuals in LS and also in QB/LB at baseline were not included in the logistic regression analysis because they did not have a space to improve. All analyses were conducted with SAS version 9.4 (SAS Institute, Cary, NC, USA).

## Ethical considerations

The Research Ethics Committee of the National Taiwan University Hospital approved this study under the annual Institutional Review Board (IRB) approval process and waived the requirement for informed consent in accordance with institutional review board regulations. This study was conducted from 2022 to 2024 under ethical approval numbers 202204036W, 202306056W, and 202403087W for each respective year. The study protocol was reviewed and approved by the Health Promotion Administration of the Taiwanese government, adhering to the ethical principles of the Declaration of Helsinki and its amendments.

## Results

### Oral habit changes distribution

There were 2,569,920 participants aged 30 years and older had at least two times of screening between 2010 and 2021. Among the participants who underwent repeated screening, 25.3% made a change to improve their oral habits. For those never chewed betel quids, participants with medium-degree and high-degree smoking changed to improving habits by 8.8% and 33.1%, respectively. Participants stopped chewing betel quids with medium-degree smoking, and high-degree smoking changed to improve habits by 24.1% and 46.1%, respectively. Low-degree chewing betel quids with no smoking, low-degree smoking, medium-degree smoking, and high-degree smoking all improved to 34.4%, 52.8%, and 69.9%, respectively. Medium-degree chewing betel quids with no smoking, low-degree smoking, medium-degree smoking, and high-degree smoking improved to 42.3%, 53.3%, and 74.2%, respectively. High-degree chewing betel quids with no smoking, low-degree smoking, medium-degree smoking, and high-degree smoking improved to 48.7%, 55.9%, and 72.7%, respectively. Males changed to improving habits at 26.2%, while females were at 20.8%. Participants aged 30–45 years, 46–60 years, and over 61 years changed to improving habits at 25.2%, 26.8%, and 23%, respectively. Among participants, 25.5% of those whose highest education level was elementary school demonstrated habit improvement, compared to 26.7% of those with education up to middle school or high school, and 22.6% of those with education beyond college. Among participants, 35.9% of those with a positive OPMD screening result showed habit improvement, compared to 24.5% in the negative screening group. Regarding participation frequency, 23.7% of those who participated only twice and 27.1% of those who participated more than twice in the oral screening program demonstrated habit improvement Table 1.

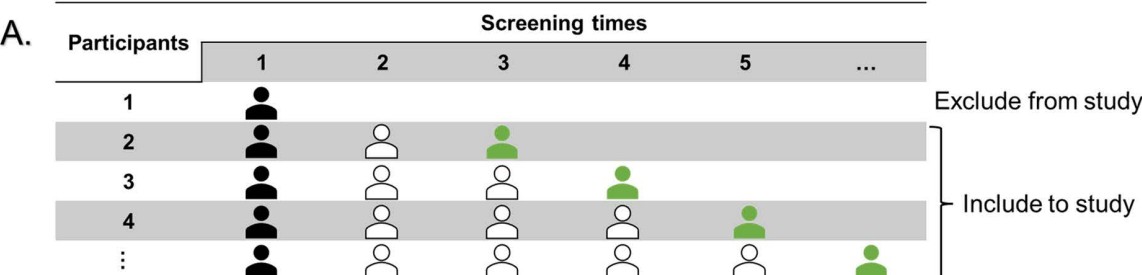

**A.**

| Participants | Screening times | | | | | |
|---|---|---|---|---|---|---|
| | **1** | **2** | **3** | **4** | **5** | **…** |
| 1 | ● | | | | | | Exclude from study |
| 2 | ● | ○ | ● | | | |
| 3 | ● | ○ | ○ | ● | | |
| 4 | ● | ○ | ○ | ○ | ● | |
| ⋮ | ● | ○ | ○ | ○ | ○ | ● |

Include to study

👤 (black) The first behavior and other characteristics were collected at this time.
👤 (green) The post behavior was collected.

**B.**

| First habit | Betel nut chewing — Post habit | | | |
|---|---|---|---|---|
| | Never chewing or Quit the chewing | Low dose chewing | Medium dose chewing | High dose chewing |
| Never chewing or Quit the chewing | | | | |
| Low dose chewing | ■ | | | |
| Medium dose chewing | ■ | ■ | | |
| High dose chewing | ■ | ■ | ■ | |

■ The improving oral habit changes

**C.**

| First habit | Smoking — Post habit | | |
|---|---|---|---|
| | Never or quit or low dose smoking | Medium dose smoking | High dose smoking |
| Never or quit or low dose smoking | | | |
| Medium dose smoking | ■ | | |
| High dose smoking | ■ | ■ | |

■ The improving oral habit changes

**D.**

| First oral habit | Post oral habit | | | | | | | | | | | | | | |
|---|---|---|---|---|---|---|---|---|---|---|---|---|---|---|---|
| | NBLS | NBMS | NBHS | QBLS | QBMS | QBHS | LBLS | LBMS | LBHS | MBLS | MBMS | MBHS | HBLS | HBMS | HBHS |
| NBLS | | | | | | | | | | | | | | | |
| NBMS | ■ | | | | | | | | | | | | | | |
| NBHS | ■ | ■ | | | | | | | | | | | | | |
| QBLS | | | | | | | | | | | | | | | |
| QBMS | ■ | | | ■ | | | | | | | | | | | |
| QBHS | ■ | ■ | | ■ | | | | | | | | | | | |
| LBLS | ■ | | | ■ | | | | | | | | | | | |
| LBMS | ■ | ■ | | ■ | | | ■ | | | | | | | | |
| LBHS | ■ | ■ | ■ | ■ | ■ | | ■ | ■ | | | | | | | |
| MBLS | ■ | | | ■ | | | ■ | | | | | | | | |
| MBMS | ■ | ■ | | ■ | | | ■ | | | ■ | | | | | |
| MBHS | ■ | ■ | ■ | ■ | | ■ | ■ | ■ | | ■ | ■ | | | | |
| HBLS | ■ | | | ■ | | | ■ | | | ■ | | | | | |
| HBMS | ■ | ■ | | ■ | | | ■ | ■ | | ■ | | | ■ | | |
| HBHS | ■ | ■ | ■ | ■ | ■ | ■ | ■ | ■ | ■ | ■ | ■ | ■ | ■ | ■ | |

■ The positive of oral habit changes

**Fig 1. Participant Criteria and Definition of Behavioral Improvement in Oral Cancer Screening.** (A.) Participants who attended at least two screening visits were included in this study. Oral habit changes were compared between the first and last screening visits, regardless of the total number

of visits, if more than two were attended. (B.) The improving oral habit changes of betel quid chewing. (C.) The improving oral habit changes of smoking. (D.) The improvement of combined oral habit changes between cigarette smoking and betel quid chewing indicates that at least 1 habit has to be improved, with no other habit getting worse. Oral habits: NB-never chewing betel quid, QB-quit the chewing, LB-low degree chewing, MB-medium degree chewing, HB-high degree chewing, LS-never or quit or low degree smoking, MS-medium degree smoking, and HS-high degree smoking.

**Table 1. Characteristics of oral habit changes.**

| Characteristics | | Subject number | Remain | Improve | Worse | P-value [a] |
|---|---|---|---|---|---|---|
| | | n | % | % | % | |
| Overall | | 2,569,920 | 34 | 25.3 | 40.7 | |
| Oral habits of the first screening | | | | | | <0.001 |
| NB | LS | 238,209 | 24.2 | 0 | 75.8 | |
| | MS | 548,007 | 45.4 | 8.8 | 45.9 | |
| | HS | 294,248 | 29.7 | 33.1 | 37.2 | |
| QB | LS | 535,669 | 66.5 | 0 | 33.5 | |
| | MS | 195,126 | 21.6 | 24.1 | 54.3 | |
| | HS | 159,444 | 23.0 | 46.1 | 30.9 | |
| LB | LS | 102,139 | 10.4 | 34.4 | 55.2 | |
| | MS | 59,682 | 6.5 | 52.8 | 40.6 | |
| | HS | 45,562 | 6.2 | 69.9 | 23.9 | |
| MB | LS | 53,000 | 22.5 | 42.3 | 35.2 | |
| | MS | 113,350 | 14.8 | 53.3 | 31.9 | |
| | HS | 87,619 | 13.8 | 74.2 | 12 | |
| HB | LS | 22,588 | 23.9 | 48.7 | 27.4 | |
| | MS | 19,825 | 12.4 | 55.9 | 31.6 | |
| | HS | 95,452 | 27.3 | 72.7 | 0 | |
| Sex | | | | | | <0.001 |
| Female | | 386,310 | 45.8 | 20.8 | 33.4 | |
| Male | | 2,183,610 | 31.9 | 26.2 | 42 | |
| Screening times | | | | | | <0.001 |
| Only 2 times | | 1,311,355 | 37.2 | 23.7 | 39.1 | |
| >2 times | | 1,258,565 | 30.7 | 27.1 | 42.3 | |
| Age groups (years) | | | | | | <0.001 |
| 30-45 | | 1,110,923 | 30.5 | 25.2 | 44.3 | |
| 46-60 | | 926,731 | 33.8 | 26.8 | 39.3 | |
| 61+ | | 532,266 | 41.5 | 23.0 | 35.5 | |
| Education | | | | | | <0.001 |
| Elementary school | | 423,075 | 36.9 | 25.5 | 37.6 | |
| Middle and high school | | 1,129,494 | 30.7 | 26.7 | 42.6 | |
| College+ | | 468,617 | 36.5 | 22.6 | 40.8 | |
| Unknown | | 548,734 | 36.4 | 24.7 | 38.9 | |
| OPMDs Screening Results | | | | | | <0.001 |
| Positive | | 191,795 | 26.9 | 35.9 | 37.2 | |
| Negative | | 2,378,125 | 34.6 | 24.5 | 40.9 | |

Oral habits: NB-never chewing betel quid, QB-quit the chewing, LB-low degree chewing, MB-medium degree chewing, HB-high degree chewing, LS-never or quit or low degree smoking, MS-medium degree smoking, and HS-high degree smoking.

[a]p values for differences in improvement were calculated using the Chi-square test and Fisher's Exact Test, as appropriate.

S1 Table in S1 File shows medium-degree betel quid chewing with high-degree smoking (MBHS) and high-degree betel quid chewing with high-degree smoking (HBHS) in males could improve habits by more than 70%, while low-degree betel quid chewing with high-degree smoking (LBHS), medium-degree betel quid chewing with high-degree smoking (MBHS), and high-degree betel quid chewing with high-degree smoking (HBHS) in females could improve habits by more than 70%. Positive OPMDs who engaged in medium-degree betel quid chewing with high-degree smoking (MBHS) and high-degree betel quid chewing with high-degree smoking (HBHS) saw an improvement in their habits of more than 70%. On the other hand, when the negative group engaged in low-degree betel quid chewing with high-degree smoking (LBHS), medium-degree betel quid chewing with high-degree smoking (MBHS), and high-degree betel quid chewing with high-degree smoking (HBHS), their habits improved by more than 70%. Improving habits more than 70% found in medium-degree betel quid chewing with high-degree smoking (MBHS) of aged 30–45 years; low-degree betel quid chewing with high-degree smoking (LBHS), medium-degree betel quid chewing with high-degree smoking (MBHS), and high-degree betel quid chewing with high-degree smoking (HBHS) of aged 46–60 years; low-degree betel quid chewing with smoking over medium-degree level (LBMS LBHS), medium-degree betel quid chewing with high-degree smoking (MBHS) and high-degree smoking with smoking over medium-degree level (HBMS HBHS) of aged over 61 years. Improving habit of more than 70% was found in low-degree betel quid chewing with high-degree smoking (LBHS), medium-degree betel quid chewing with high-degree smoking (MBHS) and high-degree betel quid chewing with high-degree smoking (HBHS) of participants whose highest education was elementary school; medium-degree betel quid chewing with high-degree smoking (MBHS) and high-degree betel quid chewing with high-degree smoking (HBHS) of participants whose highest education was middle and high school; low-degree betel quid chewing with high-degree smoking (LBHS), medium-degree betel quid chewing with high-degree smoking (MBHS) and high-degree betel quid chewing with high-degree smoking (HBHS) of participants whose highest education level was over college. Two-time screening participation resulted in an improvement of more than 70% in medium-degree betel quid chewing with high-degree smoking (MBHS); on the other hand, more than two-time screening participation led to an improvement of more than 70% in low-degree betel quid chewing with high-degree smoking (LBHS), medium-degree betel quid chewing with high-degree smoking (MBHS), and high-degree betel quid chewing with high-degree smoking (HBHS) S1 Table in S1 File.

### Factors related to better habit changes

This logistic regression model was adjusted for age group, sex, education level, oral habit severity, screening frequency, and initial OPMD diagnosis. To simplify the categorization of oral habits severity during the first round of screenings, we consolidate them into four distinct groups: only cigarette smoking, betel quid chewing combined with mild smoking, betel quid chewing combined with moderate smoking, and betel quid chewing combined with severe smoking. This approach helps to avoid an excessive number of levels. The study revealed that the favorable results of OPMDS had a significant impact on improving habits, but only in people who had a smoking habit exclusively (adjusted odds ratio (aOR) 1.18, 95% confidence interval (CI) 1.16–1.20). However, the presence of OPMD did not lead to improvement in habits for individuals who engaged in both smoking and chewing betel quids (aOR 0.84 (95% CI 0.82–0.87) for individuals who chewed betel quid and engaged in mild smoking, aOR 0.79 (95% CI 0.77–0.81) for those who chewed Betel quid and engaged in moderate smoking, aOR 0.88 (95% CI 0.86–0.9) for those who chewed betel quid and engaged in severe smoking). Males were less likely to improve their oral habits compared to females (aOR = 0.86, 95% CI: 0.85–0.87). Compared to individuals aged 30−45 years, it demonstrated that as age increased, there was a significant increase in improving oral habit. The adjusted odds ratio (95% confidence interval) was 1.31 (1.3–1.32) for individuals aged 46–60 years and 1.71 (1.69–1.73) for those over 61 years. Those with the highest education level in middle and high school showed a decreased likelihood of habit improvement when comparing their education to the elementary school level (aOR 0.94, 95% CI 0.93–0.95), while individuals with the highest level of education beyond college have a greater chance of improving their habits (aOR 1.09, 95% CI 1.08–1.11). Participating attending more than two rounds of screening, compared to two rounds of screening, significantly increased the opportunity for habit improvement (aOR 1.046, 95% CI 1.038–1.053) Table 2.

**Table 2.** The associated factors of improving habit changes.

| Characteristics | OR | 95% CI | | aOR | 95% CI | |
|---|---|---|---|---|---|---|
| Oral habits of the first screening \| OPMD results | | | | | | |
| Only cigarette smoking | | | | | | |
| Positive OPMD | 1.18 | 1.16 | 1.2 | 1.18 | 1.16 | 1.2 |
| Negative OPMD | ref | | | ref | | |
| Betel quid chewing with mild smoking | | | | | | |
| Positive OPMD | 0.82 | 0.79 | 0.85 | 0.84 | 0.82 | 0.87 |
| Negative OPMD | ref | | | ref | | |
| Betel quid chewing with moderate smoking | | | | | | |
| Positive OPMD | 0.78 | 0.76 | 0.8 | 0.79 | 0.77 | 0.81 |
| Negative OPMD | ref | | | ref | | |
| Betel quid chewing with severe smoking | | | | | | |
| Positive OPMD | 0.88 | 0.86 | 0.9 | 0.88 | 0.86 | 0.9 |
| Negative OPMD | ref | | | ref | | |
| Sex | | | | | | |
| Male | 1.19 | 1.17 | 1.2 | 0.86 | 0.85 | 0.87 |
| Female | ref | | | ref | | |
| Age groups (years) | | | | | | |
| 30-45 | ref | | | ref | | |
| 46-60 | 1.17 | 1.16 | 1.18 | 1.31 | 1.3 | 1.32 |
| 61+ | 1.3 | 1.29 | 1.31 | 1.71 | 1.69 | 1.73 |
| Education | | | | | | |
| Elementary school | ref | | | ref | | |
| Middle and high school | 0.83 | 0.82 | 0.84 | 0.94 | 0.93 | 0.95 |
| College+ | 0.684 | 0.677 | 0.692 | 1.09 | 1.08 | 1.11 |
| Screening time | | | | | | |
| Screening 2 times | ref | | | ref | | |
| Screening more than 2 times | 1.086 | 1.08 | 1.093 | 1.046 | 1.038 | 1.053 |

## Discussion

Among participants who had never chewed betel quid, those with low degree smoking habits were more likely to exhibit worsening habits (75.8%) compared to those with higher degree smoking habits. In contrast, individuals who had previously quit betel quid chewing, indicating they had already modified at least one risky habit, showed less deterioration in low-degree smoking habits (33.5%). This suggests that smoking habits are more prone to worsening than betel quid chewing, which is consistent with previous findings [31]. Moreover, when focusing on the group with high-degree smoking habits, it was found that individuals who also engaged in betel quid chewing demonstrated greater improvement in their habits. This suggests that betel quid chewing may have a stronger influence on habit modification than smoking. US and Korean populations' studies showed that people with high-degree smoking decreased the opportunity for improving oral risk habits (smoking cessation), with an OR (95% CI) of 0.42 (0.35, 0.5) and 0.67 (0.47, 0.95), respectively [25,26]. It is possible that the easier accessibility of cigarettes nowadays, for example, their availability in convenience stores, may contribute to the difficulty in improving smoking behavior.

Those diagnosed with OPMDs can increase health awareness and promote habit modification among cigarette smokers (aOR 1.18, 95%CI: 1.16–2). The study in China and Korea, which found that participant illness influenced smoking improvement [26,32], supports this result. According to a study in China, illness was the 47.3% most common

reason for smoking cessation [32]. In Korea, a study found that illness affected smoking cessation with an OR (95% CI) of 1.4 (1.07, 1.85) for hypertension and 1.68 (1.03, 2.75) for cardiovascular disease [26]. Some suspected abnormal findings on CT screening affected smoking's improvement [33,34]. Although the presence of OPMD may help motivate habit change among individuals with only a smoking habit (low-risk group), its effect appears limited among those with both smoking and betel quid chewing habits (high-risk group). This is despite the high-risk group demonstrating a higher proportion of habit improvement. These findings suggest that awareness of the dangers associated with OPMD may still be insufficient. Therefore, enhancing the effectiveness of oral cancer screening programs may require the integration of targeted strategies aimed at increasing participants' understanding of the severity and long-term consequences of OPMDs.

Male individuals demonstrated a greater capacity for enhancing their habits in comparison to their female counterparts. However, consideration of other variables revealed that males showed less improvement in their habits than females (aOR 0.86, 95%CI: 0.85–0.87). Previous research conducted in the US and China also found that men were less likely to quit smoking, with adjusted odds ratios (aOR) of 0.96 (95% CI: 0.84–1.11) in the US [25] and 0.81 (95% CI: 0.43–1.54) in China [27]. Conversely, the Malaysian study found a hazard rate ratio (HRR) of 0.82 for females who stopped chewing betel quid, with a 95% confidence interval spanning from 0.5 to 1.3 [21]. Nevertheless, the outcomes of all three trials did not demonstrate statistical significance. In China, further research showed a similar distribution, with 16.6% of males and 16.3% of females intending to quit smoking [32]. Evidence indicates that both males and females can contribute to habit enhancement, with differing effects depending on geographical area.

The opportunity for improving the oral risk habits of both models in this study is higher as age increases (aOR 1.31, 95%CI:1.3–1.32 for age 46−60 and aOR 1.71, 95%CI:1.69–1.73). However, researchers found that elderly Malaysians were less likely to stop their betel quid chewing habit (HRR 0.2, 95% CI 0.1–0.6) [21]. Additionally, some studies of smoking cessation in China and Korea showed a non-significant effect of age, with an OR (95% CI) of 0.94 (0.4, 2.23) and 2.35 (0.48, 11.45), respectively [24,27]. The disparities in oral health outcomes among regions can be attributed to differences in health awareness and adherence to traditional practices. Nevertheless, it is encouraging to see that middle-aged and older adults in Taiwan are becoming more attentive to their oral health. Even greater benefits, however, could be realized if the avoidance of risky behaviors were instilled from a younger age.

The participants' highest level of education, particularly a college degree or higher, can positively influence habit improvement (aOR 1.09, 95%CI: 1.08–1.11). The findings of this study align with prior research conducted in China and Korea. In China, a study found that education has a non-significant effect on improving cigarette smoking, with an OR (95% CI) of 1.21 (0.84, 1.74) for medium education and 1.29 (0.83, 1.98) for high education, compared to low education [27]. However, this survey study in China was limited to urban areas, but most Chinese smokers live in rural areas [27]. The study in Korea showed that higher education increases the opportunity for improving cigarette smoking, with an OR (95% CI) of 3.19 (1.02, 9.98) [24]. Highly educated persons sometimes prioritize their personal opinions over societal conventions. Once individuals comprehensively grasp the risks linked to their behaviors, they are more inclined to implement modifications.

Conducting screenings more than twice could improve participants' habits (aOR 1.046, 95%CI: 1.038–1.053). Repeated participation indicates a deliberate effort to change these oral risky habits and maintain a healthy lifestyle. Other studies in China and Korea also found the same pattern: participants who intended to quit smoking were more successful [24,26,27]. The study in Korea used the clinic visit period to demonstrate the intention to quit smoking and found the impact to be an OR (95% CI) of 7.16 (5.57, 9.2) [26]. Among Korean participants who consulted about smoking cessation, counseling increased the opportunity for success, with an OR (95% CI) of 1.87 (1.21, 2.89) [24]. Among Chinese participants, those who tried to quit smoking had more improvement in smoking cessation, with an OR (95% CI) of 2.29 (1.81, 2.89) [27]. It is anticipated that increased promotion and public awareness campaigns about the screening program could enhance participation and lead to greater improvements in oral health behaviors.

 

The investigation reveals that a quarter of the participants in an oral cancer screening program were able to effectively enhance their practices. The screening program offers both early diagnosis of abnormal lesions or diseases and knowledge about oral risk factors that can lead to oral cancer. The study found that individuals who engaged in both betel quid chewing and smoking were more likely to improve their habits than those who smoked only. Oral potentially malignant disorders (OPMDs) are conditions that may progress to oral cancer; thus, detecting such abnormalities should encourage individuals to adopt healthier behaviors. However, the positive impact of OPMD was observed only among participants who smoked but did not chew betel quid, indicating a lack of awareness about the dangers associated with OPMDs. Age-related vulnerability to illness may explain why older adults in this study appeared to have greater awareness of health risks and were more likely to engage in positive habit modification. Additionally, individuals with higher education levels may be more inclined to adhere to their own reasoning rather than societal values. Once they understand the dangers of their personal habits, they are more likely to make informed decisions to change. Regular participation in screening programs also reflects a strong and consistent commitment to personal health, which can foster successful long-term habit improvement. In Taiwan, oral cancer screening programs are conducted by trained oral health professionals, including doctors, nurses, and technicians, which enhances the credibility and effectiveness of the program in promoting healthy behaviors [35–40].

However, 75% of participants did not show improvement in their risk habits, particularly among those with low-intensity habits. It is possible that these individuals perceive low levels of consumption as harmless. If this is the case, increasing the intensity of educational content to emphasize the dangers associated with even low-level use, along with promoting more consistent participation in the screening program, may be necessary to enhance the effectiveness of habit change. Furthermore, other underlying factors may hinder improvement in this group. Future research should consider exploring additional determinants such as socioeconomic constraints and the accessibility of cigarettes and betel quid.

### Limitations

Nevertheless, we used participant cognition to gather data on their highest level of education and information about cigarette or betel quid consumption. As a result, the occurrence of recall bias is a very plausible scenario. The current study had one limitation. As we had a particular set of people who participated in the screening program at least twice as the subject of the current study, the results could not be generalized to people who participated in the screening program only once.

### Conclusion

This study highlights that 25% of participants improved their oral habits, reflecting a positive outcome of the oral cancer screening program. The most significant improvements were observed among elderly individuals, those with higher education levels, and participants who consistently engaged with the program. These findings underline the potential of such initiatives to foster healthier habits, particularly among groups more likely to be influenced by health education and regular monitoring. However, being diagnosed with oral potentially malignant disorders (OPMDs) has not proven to be a strong motivator for habitual change, except among individuals who smoked but did not engage in other risky habits, such as betel quid chewing. This indicates a gap in awareness among high-risk groups engaging in multiple risky habits, who seem less likely to recognize the dangers of their habits or the severity of OPMDs.

To address this issue, future efforts should focus on enhancing public knowledge and understanding through collaboration between health organizations and the screening program. By promoting greater awareness of the risks associated with smoking, betel quid chewing, and OPMDs, such partnerships could help individuals recognize the importance of changing their habits. Increased understanding could empower individuals to make more informed decisions about their health. Enhancing the effectiveness of the oral cancer screening program could further reduce the incidence of oral cancer over time and significantly lower the healthcare costs associated with its treatment. These combined efforts would not only improve individual outcomes but also strengthen the overall impact of the screening initiative on public health.

## Supporting information

**S1 File.** S1 Table Characteristics of oral habit change used in the logistic regression model (improve and not improve). S2 Table The number of study participants per number of visits. (ZIP)

## Acknowledgments

The authors provide their warm thanks and appreciation to all study participants.

## Author contributions

**Conceptualization:** Amy Ming-Fang Yen.

**Data curation:** Chiu-Wen Su.

**Formal analysis:** Pattaranan Munpolsri.

**Funding acquisition:** Yen-Yu Chou, Li-Ju Lin, Chao-Chun Wu.

**Investigation:** Hsu-Fei Yang, Tsui-Hsia Hsu.

**Methodology:** Chiu-Wen Su, Sam Li-Sheng Chen, Amy Ming-Fang Yen.

**Project administration:** Hsu-Fei Yang, Tsui-Hsia Hsu, Yen-Yu Chou, Li-Ju Lin, Chao-Chun Wu.

**Software:** Pattaranan Munpolsri.

**Supervision:** Sam Li-Sheng Chen, Amy Ming-Fang Yen.

**Validation:** Amy Ming-Fang Yen.

**Writing – original draft:** Pattaranan Munpolsri.

**Writing – review & editing:** Amy Ming-Fang Yen.

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
