## [Decision Letter · Decision Letter 0]

Dear Dr. Yen,

Thank you for submitting your manuscript to PLOS ONE. After careful consideration, we feel that it has merit but does not fully meet PLOS ONE’s publication criteria as it currently stands. Therefore, we invite you to submit a revised version of the manuscript that addresses the points raised during the review process.

We look forward to receiving your revised manuscript.

Kind regards,

Chung-Ta Chang

Academic Editor

PLOS ONE

3. Thank you for submitting your work to PLOS ONE. As you are reporting a retrospective study of medical records or archived samples, please ensure that you have discussed whether all data were fully anonymized before you accessed them and/or whether the IRB or ethics committee waived the requirement for informed consent. If patients provided informed written consent to have data from their medical records used in research, please include this information.

4. In the online submission form, you indicated that [Since the protocol has been reviewed by the Research Ethics Committee C of the National Taiwan University Hospital, the raw data supporting the conclusions of this article will be made available by the authors without undue reservation.].

Reviewers' comments:

Reviewer's Responses to Questions

**Comments to the Author**

1. Is the manuscript technically sound, and do the data support the conclusions?

Reviewer #1: Partly

Reviewer #2: Yes

Reviewer #3: Partly

Reviewer #4: Partly

2. Has the statistical analysis been performed appropriately and rigorously?

Reviewer #1: No

Reviewer #2: I Don't Know

Reviewer #3: No

Reviewer #4: Yes

3. Have the authors made all data underlying the findings in their manuscript fully available?

Reviewer #1: Yes

Reviewer #2: No

Reviewer #3: No

Reviewer #4: Yes

4. Is the manuscript presented in an intelligible fashion and written in standard English?

Reviewer #1: Yes

Reviewer #2: Yes

Reviewer #3: Yes

Reviewer #4: No

Reviewer #1: 1. The manuscript within the scope of the journal.

2. Both the quality and data presentation of this manuscript are acceptable and of great importance to surgeons, pathologists, radiologists and even patients.

3. The manuscript expands our knowledge about oral cancer.

4. The title should be revised and reduced its characters.

5. The abstract should reflect the content of the article and must be with range of 250-300 words.

6. Four to six keywords representing the main content of the article BUT not mentioned in the title.

7. More paragraphs should be incorporated to introduction about the details of oral cancer and treatment options .

8. The author( s) should pay attention to writing grammar and typing errors

9. The statements in discussion are acceptable but more paragraphs about the justification of your findings and comparison with other relevant studies.

10. Up to date references should be added to reference list and the old should be omitted.

11. The writing of reference list should be according to the journal style and layout for all references.

Good Luck

Reviewer #2: 1. Sample description:

The manuscript does not explain how the sample was formed or how the sampling was conducted. It would be important to specify how the households, cities, or blocks were selected to determine the sample.

2. Examination procedure:

Although the authors mention biases that occurred during data collection, it would be relevant to describe how the examinations to diagnose diseases were performed: where exactly they took place (inside the home or at a medical center).

3. Data collection process and survey origin:

It is indicated that this study used data from a national survey, apparently government-led, that monitors cancer and risk factors. It would be helpful to provide more detail on how data collection was conducted, whether the sampling method was sufficient to ensure national representativeness, and whether expansion factors were included to support that representativeness. Additionally, it would be important to specify what type of organization or institution is responsible for the Taiwan Oral Cancer Screening Program and to provide a brief description of this entity.

4. Training of field staff:

Regarding the preparation of the personnel who conducted data collection, it is mentioned that they were trained. It would be important to detail who provided the training, how long it lasted, whether they obtained any certification, and whether the interviewers were dentists.

5. Access to data:

The manuscript should clarify how Dr. Amy obtained access to the data. It would be relevant to detail the process she followed or the conditions she met to gain access to the dataset.

6. Health education and promotion:

The authors mention that participants received health education or promotion to quit smoking and improve their habits. It would be useful to describe how the oral health education was implemented: its duration, how it was conducted, and who was responsible for delivering it.

7. Statistical analysis:

In the statistical analysis, it would be beneficial to include a check for collinearity (e.g., by calculating the VIF) and mention it in the results section.

• Confidence intervals and methodological clarity:

Given the extremely large sample size (over 2 million participants), the narrow confidence intervals reported are understandable. However, to strengthen the transparency and robustness of the statistical analysis, I recommend that the authors include additional details such as:

• A clear description of the independence of observations and whether any clustering or nesting of data was present. If clustering existed (e.g., participants within hospitals or communities), please clarify whether adjustments (e.g., using robust standard errors or multilevel modeling) were made.

• The method used to calculate the confidence intervals (e.g., Wald method, robust estimates, or bootstrapping).

• A brief description of how key assumptions of logistic regression were assessed, particularly linearity of continuous predictors in the logit and checks for multicollinearity.

Providing this information will enhance the methodological rigor and allow readers to better evaluate the validity of the findings.

Reviewer #3: The work presented by Munpolsri et al is important as it examines the effectiveness of oral screening program in changing oral habits and identifies the factors influencing such screening programs at an individual level on a large scale (~2.6 million participants). The strength of this study relies on its large sample size. However, having a closer look at the paper and other work related to it, several questions came up:

1) The authors have a recently published work with the title "Behavior Changes for Smokers and Betel Quid Chewers Participating in the Organized Oral Mucosal Screening Between 2010 and 2021 in Taiwan"

doi: 10.3390/cancers17030397. Looking at the objectives of previous paper and the manuscript submitted herein, I was unable to point out the differences between the published paper and the current manuscript. I believe that authors need to make the differences (if present) clear, and the objectives more specific compared to their published paper.

2) For the introduction section

Further elaboration is needed in the second paragraph. For instance, explain or elaborate more on the following statements "in improving habits" line 64 and "a singular habit" line 65 and " an impact on one another " line 66.

3) For materials and methods

How many participants were excluded from this study.

The authors need to present the reason for excluding participants younger than 30 years from this study as previously mentioned "Behavior Changes for Smokers and Betel Quid Chewers Participating in the Organized Oral Mucosal Screening Between 2010 and 2021 in Taiwan".

The authors mentioned that trained dentists and physicians visually examined the oral cavities of study participants in an unknown setting. From my experience, to be able to properly and confidently detect OPMD an oral medicine specialist needs good lightning conditions. Therefore, it is very likely that some lesions went undetected. Moreover, some of normal variants or non OPMD can be confused with OPMD by general dentists. I have a concern that since the type of training and duration provided to the trained dentists and physician was not explained in the manuscript, this might have resulted in overdiagnosis or underdiagnosis of OPMD. An oral medicine specialists are sometimes consulted to distinguish between non-OPMD related lesion from an OPMD related lesion, who were not part the examination process in this study.

Were the three levels of categorization for cigarette smoking (low, medium, high) based on previous known categorizations in the literature, if yes please add citations.

Figure 1 caption, the statement in line 126 "between first and last time of screening ", I suggest to add irrespective of the visit number when more than two visits.

As a supplementary data (Table), would you please provide the number of study participants per number of visits . For example , how many had one visit only, two, three, four etc.

As patient education about oral cancer risks and prevention was part of this oral screening program, it is important to give a detailed explanation of the type of education provided for study participants in this program. This will benefit future studies interested in improving the impact of oral screening programs on oral health.

4) For results section

The paragraph at line 176 starting with " Quit betel quid chewing with medium-degree betel " , I suggest adding the specific supplementary table assigned to the data being explained in the paragraph. for example S1 table 1 for the statement/s related to that table.

In table 1, characteristics with oral habit changes, I would suggest providing p values to show if the differences in improvements were significant in different situations. I do appreciate that CI and OR were reported later in table 2, however, my understanding that reporting p values in this case is still necessary.

What is the explanation for the majority of participants with LS (75%) becoming worse after this program ?

In section 3.2, Please specify what factors did you adjust for in the logistic regression model when reporting the different findings in this paragraph.

5) In the discussion section, are there other papers that studied the change in smoking and betel quid chewing in oral screening programs? If yes please elaborate on the difference between your findings and others in this aspect. Specifically, if the percentage of participants becoming worse (rather than improve) was similar to your study or more or less.

Hope the authors find these comments helpful.

Reviewer #4: 1. English grammar should be revised.

2. Please clarify: based on Supplementary Table S1, there were no groups for NB-LS and QB-LS. Why do the scores presented in Table 1 differ from those in the supplementary file?

3. The discussion should be more in-depth, as several topics were presented without sufficient reasoning or explanation

4. Only 25% of subjects showed improvement in their oral habits. The remaining 75% should be discussed in more detail to explain why the oral cancer screening program did not effectively lead to improvements in oral risk habits.

5. Reduce the number of publications older than 10 years—currently 14 publications (46.67%)—to less than 10% of the total references.

**Do you want your identity to be public for this peer review?** For information about this choice, including consent withdrawal, please see our Privacy Policy

Reviewer #1: **Yes: ** Tahrir Aldelaimi

Reviewer #2: **Yes: ** Marcelo Armijos Briones

Reviewer #3: No

Reviewer #4: **Yes: ** Indrayadi Gunardi

---

## [Author Response · Author response to Decision Letter 1]

23 May 2025

Comment: We note your reply to our data availability query as follows:

"The individual-level raw data used in our study cannot be made publicly available due to ethical and legal restrictions, as specified in the protocol approved by the Research Ethics Committee C of the National Taiwan University Hospital. Public release of these data would risk compromising participant confidentiality. However, we have provided the aggregated numbers by screening mode and participant "

PLOS journals require authors to make the minimal data set publicly available without restriction at the time of publication. PLOS defines the "minimal data set" as consisting of the data used to reach the conclusions drawn in the manuscript with related metadata and methods, and any additional data required to replicate the reported study findings in their entirety. This includes:

- The points extracted from images for analysis

Additionally, PLOS requires that authors comply with field-specific standards for preparation, recording, and deposition of data when applicable.

Can you please confirm if the aggregated numbers you note you have provided would constitute your minimal data set?

Response:

Thank you for your follow-up regarding data availability.

We confirm that we have uploaded the new aggregate-level dataset, which contains the minimal data set used to support the findings of our study, to the Dryad repository. This dataset includes all relevant values used to generate statistical results and conclusions presented in the manuscript, such as:

• Aggregated counts used to calculate proportions and rates;

• Values behind reported means or percentages;

• Data used to build tables and figures, including habit change patterns, screening frequencies, and OPMD findings across subgroups.

• The dataset is accessible via the following: https://doi.org/10.5061/dryad.612jm64h3

or

http://datadryad.org/share/_Y6sTNL-JPFygiwCL95Q9EZaCD_wVZBrRlSoGUC64_A

We will enter this DOI into the full Data Availability Statement in the "Additional Information" section of the PLOS submission form, as required. We will also provide the URL passcode in the "Attach Files" section.

This uploaded file constitutes our minimal data set as defined by PLOS and complies with both journal policy and the ethical requirements approved by the Research Ethics Committee of the National Taiwan University Hospital.

We add this information to the manuscript: “The aggregated dataset used in this study has been publicly uploaded to the Dryad repository and is available at: https://doi.org/10.5061/dryad.612jm64h3.” [line: 114-115]

---

## [Editor Report · Decision Letter 1]

Changes in risk habits and influencing factors in the Taiwan oral cancer screening program

PONE-D-25-08033R1

Dear Dr. Yen,

We’re pleased to inform you that your manuscript has been judged scientifically suitable for publication and will be formally accepted for publication once it meets all outstanding technical requirements.

Kind regards,

Chung-Ta Chang

Academic Editor

PLOS ONE
---

## [Editor Report · Acceptance letter]

PONE-D-25-08033R1

PLOS ONE

Dear Dr. Yen,

I'm pleased to inform you that your manuscript has been deemed suitable for publication in PLOS ONE. Congratulations! Your manuscript is now being handed over to our production team.

Kind regards,

on behalf of

Dr. Chung-Ta Chang

Academic Editor

PLOS ONE